# Prolonged Heat Stress during Winter Diapause Alters the Expression of Stress-Response Genes in *Ostrinia nubilalis* (Hbn.)

**DOI:** 10.3390/ijms25063100

**Published:** 2024-03-07

**Authors:** Iva Uzelac, Miloš Avramov, Teodora Knežić, Vanja Tatić, Snežana Gošić-Dondo, Željko D. Popović

**Affiliations:** 1Department of Biology and Ecology, Faculty of Sciences, University of Novi Sad, Trg Dositeja Obradovića 2, 21000 Novi Sad, Serbia; iva.uzelac@dbe.uns.ac.rs (I.U.); milos.avramov@dbe.uns.ac.rs (M.A.); teodora.knezic@biosense.rs (T.K.); vanja.tatic@dbe.uns.ac.rs (V.T.); 2BioSense Institute, University of Novi Sad, Dr Zorana Đinđića 1, 21000 Novi Sad, Serbia; 3Maize Research Institute “Zemun Polje”, Slobodana Bajića 1, 11185 Belgrade, Serbia

**Keywords:** diapause, *Ostrinia nubilalis*, warm acclimation, heat stress, gene expression, heat shock proteins (HSPs), antioxidative defense system (ADS)

## Abstract

During diapause, a state of temporarily arrested development, insects require low winter temperatures to suppress their metabolism, conserve energy stores and acquire cold hardiness. A warmer winter could, thus, reduce diapause incidence and duration in many species, prematurely deplete their energy reserves and compromise post-diapause fitness. In this study, we investigated the combined effects of thermal stress and the diapause program on the expression of selected genes involved in antioxidant defense and heat shock response in the European corn borer *Ostrinia nubilalis*. By using qRT-PCR, it has been shown that response to chronic heat stress is characterized by raised mRNA levels of *grx* and *trx*, two important genes of the antioxidant defense system, as well as of *hsp70* and, somewhat, of *hsp90*, two major heat shock response proteins. On the other hand, the expression of *hsc70*, *hsp20.4* and *hsp20.1* was discontinuous in the latter part of diapause, or was strongly controlled by the diapause program and refractory to heat stress, as was the case for *mtn* and *fer*, genes encoding two metal storage proteins crucial for metal ion homeostasis. This is the first time that the effects of high winter temperatures have been assessed on cold-hardy diapausing larvae and pupae of this important corn pest.

## 1. Introduction

According to the latest report of the Intergovernmental Panel on Climate Change [1], we are nowadays facing unprecedented global climate changes. The global surface temperature has increased more rapidly since 1970 than in any other 50-year period over at least the last 2000 years. Since all ecosystems are locked into a trajectory of continued warming for the next several decades, regardless of any mitigating actions, it is crucial to assess the impact of climate warming on the ecophysiology of a wide range of species [2].

Insects are ectotherms and so their physiology, metabolism and development are, to a great extent, dependent on the ambient temperature [3]. In temperate and colder climates, the overwintering ability of many insect species is closely linked to diapause, a genetically programmed state of arrested development. Such an example is the European corn borer *Ostrinia nubilalis* (Hübner, 1796), whose larvae enter facultative diapause during their fifth larval instar and gradually become cold hardy, overwintering usually in corn stalks [4]. The primary cue triggering the onset of diapause is day length, since it represents a strong, noise-free sign of upcoming seasonal changes. The other major indicator of a season is temperature, which is subject to considerable variation throughout the years.

In recent decades, global climate changes have led to an increase in average annual temperatures and have often caused milder winters, in comparison to previous decades. However, protective mechanisms important for acquiring cold resistance in many overwintering insects develop only after gradual exposure to low temperatures during the early stages of diapause, as it happens in *Ostrinia nubilalis* [5] and *Pyrrhocoris apterus* [6].

To our knowledge there are few studies investigating how climate change, and more often above average winter temperatures, may impact successful progression through the sequential phases of diapause for these cold-adapted species. Higher temperatures experienced during the photoperiodic induction of diapause are likely to reduce diapause incidence and duration in many species. For example, blow fly *Calliphora vicina* adults reared at 20 °C produce fewer diapausing offspring, and have a shorter diapausing period, compared to adults reared at 15 °C [7]. Similarly, diapause in two closely related moth species, *Ostrinia nubilalis* and *Sesamia nonagrioides*, is prevented by high temperatures, regardless of the photoperiod [8,9]. Lastly, a recent study on the gynes of paper wasps, *Polistes dominula* and *Polistes gallicus*, has showed that warmer habitats in winter increase hibernation energy costs [10].

High temperatures could not only disrupt the normal course of the diapause program, but also lead to an increased level of oxidative stress and cause heat shock. Consequently, this would trigger changes at the molecular level, primarily in the expression of genes and proteins involved in the general stress response. Among them, the most prominent are proteins that play key roles in maintaining intracellular redox balance and protecting cells from oxidative damage, as well as those that are crucial for maintaining metal ion homeostasis [11,12,13].

Within the large family of structurally and functionally diverse proteins involved in metal ion metabolism, the most remarkable and extensively studied are metallothioneins and ferritins. Their functions are the maintenance of homeostasis of essential metal ions (Zn^2+^, Cu^2+^ and Fe^2+^), detoxification after increased, acute or chronic exposure to heavy metal ions, and protection against oxidative stress [14,15]. In *Drosophila melanogaster*, metalothionein genes are expressed differently during development and the abundance of these proteins can be increased by the inductive effect of various metals [16,17]; however, little is known about the influence of thermal stress on their expression, especially in the context of diapause.

In addition to proteins that protect organisms against oxidative damage, heat stress conditions also trigger the expression of a large family of genes encoding heat shock proteins (HSPs). Found in virtually all living organisms and expressed during exposure to all kinds of stressful stimuli, HSPs are pivotal in the general stress response [18]. When the synthesis, folding and assembling of other proteins are difficult, the increased expression, synthesis and accumulation of HSPs prevent the accumulation of non-functional proteins and their aggregates, increasing the chance for survival and faster recovery of the cell. In addition to their chaperone functions, HSP proteins are strongly upregulated after exposure to extreme temperatures, heavy metals, hypoxia, oxidative stress, UV radiation and starvation [19,20].

Despite the ongoing global warming and its threatening effects on living organisms, there is an evident lack of studies that assess the effects of high temperatures on the ecophysiology of cold-hardy diapausing species. Either the influence of long-term low temperatures on the development and maintenance of diapause has been investigated [21,22] or insects during diapause have been exposed to heat stress only for a short period of time, no longer than 24 h [23].

Thus, the aim of this study on *O. nubilalis* was to investigate the impact of prolonged heat stress during larval diapause on the expression of stress-related genes that encode proteins involved in the maintenance of redox homeostasis and in heat shock response: thioredoxin (*trx*), glutaredoxin (*grx*), ferritin (*fer*), metallothionein (*mtn*) and five HSP genes: *hsp90*, *hsp70*, *hsc70*, *hsp20.1* and *hsp20.4*. We assumed that prolonged high winter temperatures would lead to higher larval metabolic rate, which in turn, would increase oxidative and protein stress, and lead to the elevated expression of the aforementioned genes, in comparison to non-stressed diapausing and non-diapausing individuals.

## 2. Results

### 2.1. ADS-Related Genes

The results of qRT-PCR analysis revealed that transcript levels of most of the selected stress-related genes were significantly altered throughout the course of *O. nubilalis* diapause, not only as a consequence of different thermal conditions, but also as a part of the diapause program. The most striking example of this can be seen through changes in the expression of two antioxidant defense system genes, glutaredoxin (*grx*) and thioredoxin (*trx*). A strong and sharp increase in the mRNA levels of *grx* and *trx* in the middle of diapause (January) was recorded in warm-acclimated groups, where the expression of these genes was 2-(*trx*) up to 5-fold (*grx*) higher than that of the non-diapausing (nd) control group (Figure 1a,b). Contrary to this, in diapausing larvae held in field conditions, the expression of *grx* and *trx* was more or less stable, close to the level of the non-diapausing control, except at the end of diapause, when an observable decline in mRNA levels was recorded for both genes (Figure 1a,b).

On the other hand, two genes crucial for maintaining metal ion homeostasis, ferritin (*fer*) and metallothionein (*mtn*), are consistently and strongly upregulated throughout the diapause period, independent of the temperature regime (Figure 2a,b). However, their expression in both field- and warm-acclimated groups has a decreasing trend, being highest at the very beginning of diapause and reaching its minimum in April, except for *mtn*, where this decline was much slower in warm-acclimated groups and the expression was even raised towards the end of diapause (April) (Figure 2a,b).

### 2.2. Heat Shock Protein Genes

When it comes to the heat shock protein genes, after the initiation of diapause, three of the five HSP genes were upregulated (*hsp90*, *hsp70* and *hsp20.4*), one was downregulated (*hsc70*), while the expression of *hsp20.1* was stable during almost the entire diapausing period, regardless of thermal conditions (Figure 3 and Figure 4). Further progression through the sequential phases of diapause did not alter the expression patterns of *hsp20.4*, which remained constantly and strongly upregulated (around 2-fold) until the end of diapause, irrespective of thermal stress (Figure 4b). The mRNA level of *hsp70* also remained observably high in warm-acclimated larvae, compared to those maintained under field conditions, whose expression showed a gradual decline from the onset of diapause to its very end (much the same as *fer* and *mtn* described above) (Figure 3b). Similar to *hsp20.4* and *hsp70*, the expression of *hsp90* was upregulated during mid-diapause (January) in larvae from both field and warm conditions. Then, throughout the latter part of diapause, the expression level of *hsp90* remained at a level similar to that of the non-diapausing group until metamorphosis to pupae, when a strong augmentation of *hsp90* mRNA was observed (Figure 3a).

The only gene that has shown downregulation at the beginning of diapause was *hsc70*, whose transcriptional level then constantly and gradually increased during the maintenance phase of diapause, first reaching the level of the non-diapausing control, after which it kept rising until diapause termination. The highest level of *hsc70* expression (almost 10-fold higher in comparison to December) was recorded in field-acclimated pupae, and in warm-acclimated larvae sampled in March (3-fold higher) (Figure 3c).

In contrast to the trends described previously, expression of *hsp20.1* during early and mid-diapause was not induced by diapause development or temperature treatment. A significant increase in *hsp20.1* expression was observed only in March in field conditions, followed by a 7-fold peak in mRNA level in pupae that emerged in April (Figure 4a). In warm-acclimated groups, there was no difference in *hsp20.1* expression throughout diapause.

At the pupal stage, the expression of *fer* and *mtn*, as well as of almost all investigated HSP genes in both non-diapausing pupae (ndp) and pupae emerged from diapausing larvae in April (p_f_) was observably enhanced (up to 10-fold) compared to those of non-diapausing larvae. However, there are few exceptions—*grx*, *trx* and *hsc70*, where no significant difference between non-diapausing larvae and pupae was recorded.

In order to assess the separate and combined effects of temperature acclimation and diapause time course on the relative gene expression, two-way ANOVA was performed using the paired data for diapausing groups from November to April, excluding non-diapausing larvae (nd) and both groups of pupae (ndp and p_f_). The results are presented in the upper left corner of each relative expression graph (Figure 1, Figure 2, Figure 3 and Figure 4). As can be seen in Figure 1 and Figure 2, the relative expression of all genes encoding ADS proteins was significantly altered as a consequence of both temperature acclimation and diapause time course, as well as interaction between the two. Similar results were observed for HSP genes (Figure 3 and Figure 4), with the exception of *hsp90*, where neither temperature acclimation nor interaction of diapause time course and temperature acclimation modulated the relative expression (Figure 3a).

### 2.3. Principal Component Analysis (PCA)

Data were also subjected to principal component analysis (PCA) in order to uncover general patterns of gene expression changes in response to temperature acclimation and time course of diapause. The first two principal components extracted by PCA accounted for 60.75% of data variance (Figure 5).

The first principal component (PC1) accounted for 39.64% of the total variance and it is highly likely that it separated samples according to developmental stage, since PC1 separated non-diapausing groups—nd, ndp and p_f_ (negative scores) from all other field- (f) and warm-acclimated (w) diapausing groups (nov, d_f_, d_w_, j_f_, j_w_, f_f_, f_w_, m_f_ and m_w_—the first letter denotes month (November to March), the second letter in subscript denotes acclimation condition; positive scores) (Figure 5a). Interestingly, PC1 sorted diapausing larvae sampled in April (a_f_ and a_w_) from non-diapausing ones, probably because their metabolism, just ahead of metamorphosis, resembles that of active forms. In addition, PC1 was positively correlated with the relative expression of most of the genes, except three heat shock protein genes—*hsp90*, *hsc70* and *hsp20.1—*which had shown a negative correlation (Figure 5b).

The second principal component (PC2), although less substantial, accounted for 21.11% of the total variance. To some extent, PC2 might reflect the effect of acclimation temperature on relative gene expression, since it partly separated warm-acclimated (d_w_, j_w_, f_w_, a_w_; positive scores) from the field-acclimated groups (d_f_, j_f_, f_f_, a_f_; negative scores) (Figure 5a). Although there are certain deviations (e.g., for the larvae sampled in March), the formation of certain data clusters above and below zero is observed for warm- and field-acclimated groups, respectively. This is supported by the fact that PC2 had a strong positive correlation with the relative expression of all analyzed genes (Figure 5b).

As is summarized in the heat map (Figure 6), the most remarkable changes in gene expression were observed for *hsp70* and *hsp20.4*, as well as for *fer* and *mtn*, all of them being strongly upregulated during diapause, in comparison to non-diapausing larvae. On the other hand, the only gene whose expression was at low level at the beginning of diapause and then sharply increased towards its end was *hsc70* (Figure 6).

## 3. Discussion

Almost all insects of temperate and polar climates employ diapause as a strategy for surviving long unfavorable periods of life. Even though the molecular processes behind cryoprotective mechanisms and increased tolerance to low temperature during diapause of the European corn borer *Ostrinia nubilalis* are well documented [5,24,25,26,27,28,29,30], little is known about the potential impact of unusually high winter temperatures on the ecophysiology of this cold-hardy insect species. Considering that both qualitative and quantitative changes in gene expression are likely to play a critical role in acclimation and adaptation to various environmental stressors, in this study we investigated the combined effects of thermal stress and the diapause program on the expression of selected genes involved in antioxidant defense and heat shock response. This pioneer study is the first of its kind to tackle the issue of ongoing climate changes on the ecophysiology of this important corn pest.

As is known, diapause involves the silencing of many genes, but a small number of genes are uniquely expressed at this time [11]. When we take a closer look, it can be observed that gene expression patterns are complex: certain genes are expressed only in early diapause, others late in diapause; some are continuously expressed, and others still are expressed intermittently [11]. However, if we add an effect of abnormally high temperatures experienced during diapause, the expression patterns become even more complex, and hard to distinguish and interpret.

Overall, in this study, three expression patterns were observed: genes that are developmentally regulated by the diapause program, genes that are predominantly heat-stress sensitive, and a group of genes that are developmentally regulated and heat-stress sensitive at the same time. The first group could consist of genes encoding the small heat shock proteins hsp20.1 and hsp20.4, as they were either diapause-regulated ahead of metamorphosis, like *hsp20.1*, or showed stably high expression that persists throughout diapause, compared to non-diapause, like *hsp20.4* (Figure 4). In addition, genes encoding two metal storage proteins, ferritin (*fer*) and metallothionein (*mtn*), would also belong to this group, since their expression was developmentally regulated rather than thermal sensitive (Figure 2). Genes encoding glutaredoxin and thioredoxin could form the second group, in which expression in warm-acclimated groups remains at a high level even in the middle diapause, when the deepest metabolic depression is expected (Figure 1). Finally, the other HSP genes: *hsp90*, *hsp70* and *hsc70* could form a third group, in which the combined effect of the diapause program and heat stress gives a unique expression pattern, different from all described so far (Figure 3).

### 3.1. Expression of ADS-Related Genes

#### 3.1.1. Thioredoxin and Glutaredoxin

Elevated expression of thioredoxin and glutaredoxin, two essential constituents of the main redox homeostasis mechanisms, was documented in numerous insects after exposure to heat stress. For example, increased expression of genes encoding thioredoxin or thioredoxin-like protein in response to high temperatures was determined in the larval fat body of the silkworm *Bombyx mori* [31] and adults of the bumblebee *Bombus ignitus* [32]. Similarly, in larvae of two lepidopteran species—the cotton bollworm *Helicoverpa armigera* and the oriental fruit moth *Grapholita molesta*—the expression levels of selected thioredoxin (*trx*) genes were strongly induced by high temperatures [33,34]. Apart from the mentioned *trx* genes, three glutaredoxin (*grx*) genes were also upregulated in *H. armigera* larvae at 12 h after 20 °C, 35 °C and 45 °C treatment [35].

In this study on *O. nubilalis*, the relative expression of two analyzed redoxin genes—glutaredoxin (*grx*) and thioredoxin (*trx*)—appears to be mostly influenced by heat stress during diapause. In particular, it can be observed in Figure 1 that these two genes become upregulated during the middle point of diapause in warm-acclimated larvae (j_w_ and f_w_ groups for *grx*, and d_w_ and j_w_ groups for *trx*) in comparison to field-acclimated larvae. Additionally, *grx* also remains upregulated towards the termination of diapause in warm-acclimated larvae (a_w_ group) when compared to diapausing larvae reared in field conditions. As such, looking at the expression patterns of these two genes in field- and warm-acclimated diapausing larvae (Figure 1a,b), it can be seen that abnormal temperatures during the resting phase of this insect’s life cycle can disturb its antioxidative defense system. Diapausing ECB larvae require exposure to ever decreasing temperatures in order to undergo changes at the level of genes and metabolism necessary for becoming cold hardy and surviving the winter [24,26,29,36,37]. Abnormally high temperatures during this period lead to a higher metabolic rate in diapausing larvae, rapid dehydration and premature exhaustion of energy reserves [38], which in turn, can trigger increased production of ROS/RNS and subsequent oxidative stress that needs to be addressed. Upregulation of ADS-related genes in these circumstances, such as *grx* and *trx*, would be one such protective mechanism.

The results obtained in this study are in accordance with similar findings on the effects of different types of stress on the expression of *grx* and *trx* in various species. The expression of *grx2* was upregulated in several larval tissues of the Asian corn borer *Ostrinia furnacalis* following exposure to high and low temperatures, UV radiation, infection with *E. coli* and mechanically induced injuries [39]. Similar results were shown for *grx1* and *grx2* from the Asian honey bee *Apis cerana cerana* [40], as well as for *grx*, *grx3* and *grx5* from the cotton bollworm *H. armigera* [35], and the Oriental fruit moth *G. molesta* [41].

Interestingly, the expression of the *trx2* gene was upregulated in adults of *A. cerana cerana* reared at 25 °C, but repressed when exposed to the 42 °C treatment [42]. This in contrast to the *trx1* gene and two glutaredoxin genes (*grx1* and *grx2*) from the same species, which were all induced when exposed to the 42 °C treatment [40]. Exposure to low temperatures led to an upregulation of *trx* in the bumblebee *B. ignitus* [32], larvae of the silkworm *B. mori* [31], as well as the sleeping chironomid *Polypedilum vanderplanki* [43] and the wood frog *Rana sylvatica* [44]. Oxidative stress also triggers the upregulation of *trx*, as observed in *A. cerana cerana* [42], and larvae of the silkworm *B. mori* [31], as well as thioredoxin-like protein genes in the bumblebee *B. ignitus* [32].

Whether the expression of *trx* and *grx* genes could be affected during different types of dormancies, such as anhydrobiosis, has also been investigated. The sleeping chironomid *P. vanderplanki* is the most complex organism able to survive desiccation and its TRX set of genes is unprecedentedly expanded to up to 25 *trx* genes, of which, virtually all have been upregulated after 24 h of desiccation [45]. Moreover, the expression of many genes related to the oxidative stress response was significantly induced during the desiccation process in this midge, with the gene encoding mitochondrial thioredoxin being the most strongly upregulated one [43]. That said, studies dealing with the expression of genes for glutaredoxin and thioredoxin after exposure to high temperatures during dormant states are rather rare. Such an experiment was conducted in diapausing larvae of *O. nubilalis* maintained at 22 °C in controlled, laboratory conditions and it was shown that the two redox-related genes had opposing expressions—*grx* was upregulated and *trx* downregulated [29]. While these results differ from the findings in the present study, the discrepancy could be owing to the experimental design. In the present study, diapausing ECB larvae were left inside intact corn stalks and exposed to the ambient conditions during the whole time course of diapause, while in the study of Popović et al. [29], the larvae were exposed to a controlled, stable temperature, reared in incubators.

#### 3.1.2. Metallothionein and Ferritin

Overall, the relative expression of two genes coding metal storage proteins, *mtn* and *fer*, was higher in diapausing larvae compared to non-diapausing larvae, regardless of acclimation conditions (Figure 2). Notable exceptions are the d_f_ group, where *mtn* expression was the lowest out of the entire experimental setup, and the a_f_ group, where relative expression of *fer* was the lowest from both setups of diapausing larvae. Additionally, relative expression of the two genes was higher in both groups of pupae in comparison to non-diapausing larvae (Figure 2a,b). Discounting the d_f_ group, relative expression of *mtn* and *fer* appears to be governed by the overall diapausing program, rather than acclimation conditions, as there is a similar expression pattern throughout most of the diapause. The sharp drop in *mtn* expression recorded in the d_f_ group is likely the result of metabolic perturbations that occur in *O. nubilalis* larvae exposed to low temperatures in early diapause, as part of their cold hardening process. Similar results to these were also found in *Sarcophaga crassipalpis*, where *mtn* was upregulated in diapausing larvae [46].

Different types of stress can also lead to increased expression of *mtn*, such as anoxia [44], cold exposure [47] and ingestion of heavy metals [48,49,50]. When it comes to *fer*, increased expression of this gene was recorded during diapause in a number of different species and during different life stages, such as in the marine copepod *Calanus finmarchicus* [51], the parasitic wasp *Nasonia vitripennis* [52], embryos of *Artemia franciscana* [53], the cotton bollworm *H. armigera* [54] and the flesh fly *S. crassipalpis* [46]. As with *mtn*, expression of *fer* is also responsive to different types of stress, such as dehydration and anoxia [55,56,57]. Similarly to *grx* and *trx*, these two genes have also been the subject of another study on the effects of above-average temperatures during the diapause of *O. nubilalis*. Exposure of larvae to 22 °C during the early months of diapause strongly induced the expression of *mtn* and *fer*, indicating their stress responsiveness [29]. This is in contrast to the findings of the present study, at first glance. However, and this bears repeating, larvae in the aforementioned study by Popović et al. [29] were directly exposed to the temperature conditions and treatments, as opposed to the diapausing larvae in our study that were more insulated by being in intact corn stalks, tuning their response to exposure to high temperatures. Regardless, it can be seen that both *mtn* and *fer* can be upregulated as a result of environmental stress. During diapause, stress can be caused by different factors such as thermal fluctuations, overall dehydration and the increased generation of ROS/RNS [12,13,58,59]. In particular, diapausing larvae experience between a 20 and 30% loss of total body water during this resting period [60], increasing the concentration of, amongst other things, metal ions that can have a profoundly toxic effect on the organism. In normometabolic conditions, such as in non-diapausing larvae and pupae, these ions would serve as cofactors for various enzymes and transcription factors. However, metabolic and transcription rates are suppressed as part of the ongoing diapause program, decreasing the consumption of metallic ions as cofactors that then need to be stored and sequestered in order to prevent them from becoming toxic to the larvae. One such solution would be the upregulation of metal storage proteins, such as metallothioneins, to counter this concentration increase in free metal ions, as these proteins readily bind common heavy-metal-based cofactors such as zinc or copper [61,62]. In addition, dormant organisms, such as the diapausing larvae of the European corn borer, are prone to infections by bacterial pathogens that require iron for growth and development. The upregulation of ferritin, the iron-binding protein, would mediate the removal of this metal and lessen the chance for bacterial infections to develop during dormancy [29,63].

### 3.2. Expression of HSP Genes

#### 3.2.1. ATP-Dependent HSPs—HSC70, HSP70 and HSP90

There are countless examples of the upregulation of HSP genes and proteins under heat stress in insects. In the fruit fly *Drosophila mojavensis*, one third of its entire transcriptome was affected after 12 h of exposure to temperatures of 15 °C, 25 °C and 35 °C, including significant upregulation of six genes encoding HSPs [64]. Similarly, in the whitefly *Bemisia tabaci*, the entire set of stress-related genes was thermally induced at 40 °C—among them those encoding three heat shock proteins—HSP90, HSP70 and HSP40 [65]. In addition to this, ample studies on the level of expression of different HSP genes have been conducted in several diapausing species in the order Lepidoptera. The expression of *hsp90* was augmented as part of the diapause program in species such as the bamboo borer *Omphisa fuscidentalis* [66] and rice borer *Chilo suppressalis* [67]. The same was detected in larvae of the corn stalk borer *S. nonagrioides* for *hsp90* and *hsc70*, while *hsp70* and *hsp20.8* were downregulated [68,69,70].

In this study, the most noticeable result of imposed thermal stress on the expression of selected *O. nubilalis* HSP genes was recorded for heat shock protein cognate 70 (*hsc70*), whose transcriptional level was not only developmentally regulated, but also highly responsive to heat stress (Figure 3c). In comparison to non-diapausing control, the expression of *hsc70* has been strongly silenced during the initial and middle stages of diapause, in accordance with general metabolic suppression related to low temperatures. Towards the end of diapause, the accruement of *hsc70* transcripts was observed for both field- and warm-acclimated groups, reaching its peak in the p_f_ group, as a part of metabolic reactivation, i.e., increased anabolism ahead of and during metamorphosis. Described downregulation of *hsc70* at the beginning of diapause was observably less pronounced in warm-acclimated larvae, where thermal stress caused a significantly higher expression of *hsc70* in comparison to those from field conditions during almost the entire course of diapause (Figure 3c). This was not the case in larvae only from April, whose level of *hsc70* falls under heat stressed and simultaneously raises in the non-stressed group, probably as a part of upcoming metamorphosis.

Rinehart et al. [71,72] proposed that *hsc70* belongs to a group of genes that are constitutively expressed and uninfluenced by pupal diapause in *S. crassipalpis*. This has been confirmed in other similar studies: in larval diapause of the rice stem borer *C. suppresalis* [67], and in pupal diapause of the solitary bee *Megachile rotundata* [73] and the corn earworm *Helicoverpa zea* [74]. However, our results are more consistent with the studies evidencing that *hsc70* has a variable pattern of expression throughout diapause—while it was low during prediapause and the first half of diapause, its expression increased greatly in the second half before diapause termination, the same as in the earlier study of diapause in *O. nubilalis* [29], as well as in several other species: the bamboo borer *O. fuscidentalis* [66], the wheat blossom midge *Sitodiplosis mosellana* [75], the spotted lanternfly *Lycorma delicatula* [76], the mosquito *Culex pipiens* [77] and the bumblebee *Bombus terrestris* [78]. Cheng et al. [75] even suggested that this gene may serve as a molecular marker to predict diapause termination. The only evidence of augmentation of *hsc70* during diapause is the paper of Gkouvitsas et al. [68], who proposed that this gene is induced during deep larval diapause in the corn stalk borer *S. nonagrioides*.

When the mRNA level of *hsc70* was examined in the context of thermal stress, the results were not consistent with each other. While heat shocking produced only a slight increase in *hsc70* expression over control background levels in *M. rotundata* [73] or the expression was constitutive, as observed for *S. nonagrioides* [68], in *O. nubilalis*, this gene was strongly induced after two months of diapause at 22 °C [29].

In the case of the other two ATP-driven heat shock protein genes analyzed in this study—*hsp90* and *hsp70*, no significant difference in *hsp90* expression, as a function of temperature, was observed until late diapause. Meanwhile, the mRNA level of *hsp70* was significantly higher in the warm-acclimated groups during mid- and late diapause in comparison to those that were field-acclimated (Figure 3a,b). This could be explained by the fact that HSP90 is more hormonally controlled and thus accrues as diapause terminates, in contrast to HSP70, which is the one that first interacts with small ATP-independent HSP proteins (sHSPs), binding to unfolded or partially unfolded proteins to prevent their aggregation and to release them for refolding [79]. Additionally, HSP70 potentially combines with compromised proteins when ATP is limited, thereby augmenting protein storage by sHSPs and other HSPs during diapause-dependent metabolic suppression [18].

As is currently known, expression of *hsp90* appears to be controlled by 20-hydroxyecdysone [80]: *hsp90* is upregulated when ecdysteroids are present, since this protein binds to the ecdysone receptor and is required for its activity in vivo. In this paper, we have shown that in field-acclimated larvae, the level of *hsp90* mRNA remained more or less stable during diapause, close to the level of the non-diapausing control, until the termination of diapause, when a strong augmentation of *hsp90* was recorded at the pupal stage (Figure 3a). This is in accordance with numerous studies in which *hsp90* expression was either indifferent [73,81,82] or downregulated during the diapause of insects belonging to the orders Diptera [52,83,84] as well as Lepidoptera [66,74,85]. However, a previous study on *O. nubilalis* suggested that *hsp90* has been markedly elevated as a part of the diapause program [29], which is consistent with patterns seen in other diapausing species: *S. nonagrioides* [69], *C. suppressalis* [67], *S. mosellana* (75), *Delia antiqua* [86], etc. The contradictory results may reflect environmental conditions experienced by the insects, the experimental methods employed and the life stage at which diapause occurs in a certain species.

When it comes to the effects of thermal stress on the transcriptional level of *hsp90*, in warm-acclimated larvae, in this study, the expression of *hsp90* was augmented during most of the diapause, compared to the non-diapausing control. This was probably because of the damaging effects of high temperatures on the protein stability and conformation. In a functional HSP network, HSP90 folds nascent proteins and adopts denaturing proteins transferred from HSP70 in a complex with cochaperone HSP40 [18]. Thus, the elevated expression of *hsp90* seems to be an integral part of an organism’s response to stressful conditions imposed by both high temperatures and diapause. This is also supported by the results of Li et al. [87] who observed that three of four heat shock protein DnaJB subfamily genes (similar to cochaperone HSP40) were upregulated by different degrees of heat stress in *A. cerana cerana*. Moreover, since larvae from the warm-acclimated group failed to reach the pupal stage, upregulation of *hsp90* at the end of diapause was not observed in this group.

The fact that thermal stress induced a rapid accumulation of *hsp90* mRNA has already been documented in other diapausing insects [83,86,88,89]; however, here, only short-term heat stress was imposed on diapausing individuals, ranging from 1 to a maximum of 24 h, so these results are not quite comparable to ours.

In contrast to *hsp90* expression, plenty of studies indicate that genes encoding proteins of the HSP70 family are strongly upregulated during the diapause of different dipteran species: *S. crassipalpis* [71,72,83,84], *Calliphora vicina* [90], *M. rotundata* (73) and *D. antiqua* [91]. There is also strong evidence that altered *hsp70* expression plays an important role in the temperature adaptation of some lepidopteran species, such as the moth *G. molesta* [92] and the butterfly *Melitaea cinxia* [93]. Sometimes two isoforms of the same HSP70 show different regulation during diapause, as described in the Colorado potato beetle *Leptinotarsa decemlineata* [94]—*hsp70a*, but not *hsp70b*, was upregulated during adult diapause, while, at the same time, both isoforms were highly induced by heat shock in non-diapausing individuals. The described gene expression pattern could also differ if the study was conducted in the tissue-specific manner, as is shown by Lebenzon et al. [95], who observed an increase in the *hsp70* transcript in the flight muscle, but not in the fat body, of diapausing *L. decemlineata*.

#### 3.2.2. ATP-Independent HSPs—HSP20.1 and HSP20.4

The expression patterns of genes encoding two small heat shock proteins in this study were completely opposite—*hsp20.1* was diapause regulated only ahead of metamorphosis, while *hsp20.4* showed a stably high expression, which persists throughout diapause, compared to the active phase. Namely, in the first months of diapause, the mRNA level of *hsp20.4* was elevated in larvae kept in field conditions, in comparison to warm-acclimated larvae. However, from February until the termination of diapause, expression was maintained at approximately the same level in both experimental groups (Figure 4b). On the other hand, no significant difference in *hsp20.1* expression, as a function of temperature, was observed. The transcriptional level of *hsp20.1* differed between stressed and non-stressed groups only in March, when a strong increase in expression was observed in field-acclimated larvae (Figure 4a).

Interestingly, diapausing larvae of the corn stalk borer *S. nonagrioides* also displayed the differential regulation of genes coding small HSPs—expression of *hsp19.5* was constant, while *hsp20.8* was downregulated in deep diapause and then upregulated at the termination of diapause [70]. In addition, a substantial difference in the expression of genes for small HSPs has been previously documented in *O. nubilalis*, where *hsp20.4* was upregulated and *hsp20.1* downregulated during the entirety of diapause, showing a sharp increase only at the very end of diapause [29].

These results could be explained in light of the cell function of the small heat shock proteins (sHSPs). Acting independently of ATP, they are the first line of cell defense, preventing the irreversible denaturation of substrate proteins, especially when cells are stressed [18]. Small HSPs have been extensively studied in diapausing insects, including the fruit flies *Drosophila triauraria* and *D. melanogaster* [23,81], the flesh fly, *S. crassipalpis* [96], the corn stalk borer *S. nonagrioides* [70], the wheat *midge S. mosellana* [97], the butterfly *Pieris melete* [98], the leaf beetle *Gastrophysa atrocyanea* [99] and the silkworm *B. mori* [100]. The augmentation of sHSPs appears to be common during diapause in numerous species across different insect orders, including Lepidoptera, Diptera, Hymenoptera and Coleoptera, and can occur at different developmental stages (embryo, larva, pupa, adult) [72]. However, some studies propose that this is not the case in all insect species studied so far [67,81,82,101]. Different expression patterns of *hsp20.1* and *hsp20.4*, observed in this study, imply that they have distinct functions during the active and passive phases of development—*hsp20.1* seems to be more metabolism connected, while *hsp20.4* is more stress protection connected.

## 4. Materials and Methods

### 4.1. Experimental Design

Corn stalks infected with larvae of the European corn borer (ECB) were obtained from the Maize Research Institute in Zemun Polje (44°87′ N, 20°33′ E), Serbia, in autumn 2017. In November 2017, early diapausing fifth instar larvae of the European corn borer (ECB) were randomly sampled from these stalks, frozen in liquid nitrogen and stored at −80 °C until analysis. These larvae were set as the control group since they represent the starting point for the heat stress treatment (Figure 7). The remainder of the corn stalks were placed in two different locations—one outdoors under field conditions, and the other indoors under controlled laboratory conditions at temperatures from 6 to 11 °C during the night and 12 to 22 °C during the day. It is important to note that the indoor ambient temperatures were significantly higher than the average field temperatures to which *O. nubilalis* is adapted during its winter diapause.

Following that, larvae were sampled simultaneously each month from corn stalks in both locations, frozen in liquid nitrogen and stored at −80 °C until analysis. A total of eleven experimental groups of diapausing larvae were established (Figure 7): one control group from larvae sampled in November (nov), and ten groups from larvae sampled from December to April, held in either field (f) or warm acclimation (w) conditions (d_f_, d_w_, j_f_, j_w_, f_f_, f_w_, m_f_, m_w_, a_f_, a_w_; first letter denotes month, second letter in subscript denotes acclimation condition). When it comes to pupae, only one group was sampled—those emerged from larvae under field conditions (p_f_) in April, since larvae of the warm-acclimated group failed to reach the pupal stage. After sampling enough pupae to form the p_f_ experimental group, the rest were placed in mating cages and left to complete their life cycle and produce a generation of non-diapausing larvae. Adult moths, emerged from pupae after metamorphosis, mated and laid eggs, which were placed in glass jars with insect feed prepared with components according to Table 1.

Larvae that emerged from the eggs were reared under non-diapausing light conditions (18 h L:6 h D) until they reached the 5th larval instar and pupal stage, when they were collected for the last two experimental groups: non-diapausing larvae (nd) and non-diapausing pupae (ndp), respectively (Figure 7). The larvae and pupae were then frozen in liquid nitrogen and stored at –80 °C until analysis. Each of the fourteen experimental groups comprised three biological pools and each pool contained three larvae or pupae. Non-diapausing larvae (nd) were set as a second control group, beside diapausing larvae from November (nov), since these two groups were not subjected to any temperature treatment.

### 4.2. Total RNA Extraction, cDNA Synthesis and qRT-PCR

Total RNA was isolated from the whole bodies of larvae or pupae from the above samples using the TRIzol Reagent (Invitrogen, Waltham, MA, USA, cat. no. 15596026) and following the manufacturer’s protocols. The purity and concentration of the RNA samples were assessed using the BioSpec-nano spectrophotometer (Shimadzu, Kyoto, Japan). To determine the integrity of the isolated RNA, samples were run on 1.5% agarose gel and the 28S/18S intensity ratio was used to calculate the RNA integrity number (RIN). First-strand complementary DNA (cDNA) was synthesized from 4 μg of total RNA using the High-Capacity cDNA Reverse Transcription Kit (Applied Biosystems, Waltham, MA, USA, cat. no. 4368814). The reverse transcription process was carried out on an Eppendorf Mastercycler EP Gradient S (Eppendorf, Hamburg, Germany) in thermal cycling conditions as follows: 25 °C for 10 min, followed by 37 °C for 120 min and 85 °C for 5 min, and hold at 4 °C indefinitely. Obtained cDNA was diluted 8 times with nuclease-free H_2_O, adjusting the concentration to 12.5 ng/μL, and immediately stored at −80 °C for later use.

The relative abundance of mRNA transcripts was assessed by quantitative real-time PCR (qRT-PCR) carried out on a CFX Connect™ Real-Time PCR Detection System (Bio-Rad, Hercules, CA, USA). Power SYBR™ Green PCR Master Mix (Applied Biosystems, Waltham, MA, USA, cat. no. 4367659) was used to set up qRT-PCR reactions, according to manufacturer’s instructions. Single reaction mixtures were prepared according to Table 2 and thermal cycling conditions were as follows: 50 °C for 2 min, then 95 °C for 10 min, followed by 40 cycles of 95 °C for 15 sec and 60 °C for 1 min. The specificity of the amplified product was further confirmed through a melting curve analysis from 60 °C to 95 °C. All samples were run as technical duplicates, while each experimental group had three biological replicates.

Nucleotide sequences of the genes of interest were collected from the NCBI database, all of them belonging to *O. nubilalis*, except for *hsp70*, which belongs to the closely related Asian corn borer *Ostrinia furnacalis*. The primer pairs for selected genes (Table 3) were designed using the Primer-BLAST tool [102], and ordered from Vivogen LLC (Belgrade, Serbia). The gene coding actin was used as the house-keeping gene for normalization.

### 4.3. Relative Expression and Statistical Analyses

Relative expression of analyzed genes was calculated according to the methodology described by Ganger et al. [103], which is derived from the Livak and Schmittgen [104] and Pfaffl [105] methods. This mathematical model keeps all calculations in the logscale by using logarithmized primer efficiency values (log_10_E); subsequent statistical analyses are then also applied in the logscale [103]. Next, mRNA expressions of the target genes are normalized by subtracting their Ct (cycle threshold) values from the Ct values of the reference gene (*actin*) and the final results are expressed in ΔCt values.

The statistical significance of the differences between ΔCt values of genes from all experimental groups was tested with one-way ANOVA and *post hoc* Fisher’s test, with a level of significance of *p* < 0.05. In order to test the combined influence of both diapause time course and acclimation conditions on the relative expression of each gene, a two-way ANOVA followed by *post hoc* Tukey’s test, with a significance level of *p* < 0.05, was applied to all experimental groups from November to April for each gene separately. Non-diapausing larvae and both groups of pupae were excluded from the two-way ANOVA analysis due to the lack of influence of different acclimation conditions on these groups in the existing experimental set up. Results were also subjected to principal component analysis (PCA) in order to reveal specific patterns of gene expression under different experimental conditions. For this analysis, we used the ΔCt values of all genes from all experimental groups and obtained two plots: a PCA score plot (Figure 5a), which shows clusters of samples based on their similarity, as well as a PCA loading plot (Figure 5b), which shows how strongly each characteristic influences a principal component.

All statistical analyses were performed using the Statistica version 14.0.0.15 software (StatSoft, Inc., Tulsa, OK, USA). Finally, in order to underline the differences in gene expression across groups of diapausing larvae held in field- and warm-acclimated conditions, a heat map of relative expression for all analyzed genes, normalized to *actin* and compared to the non-diapausing group (nd) as a control, was created in R Statistical Software version 4.3.2 (https://www.r-project.org/ accessed on 20 December 2023).

## 5. Conclusions

The results of this and previous research on the European corn borer and other diapausing insects show that low temperature exposure during diapause is crucial for the maintenance of metabolic depression and conservation of energy reserves, which enable insects to survive the long and stressful resting period that is diapause. However, due to the ongoing and apparent global warming, insects of temperate and other cold regions experience unusually high winter temperatures. Here, we have shown that prolonged heat stress during larval diapause in *O. nubilalis* induces a higher metabolic rate and oxidative stress, and thus reflects on their ability to terminate diapause and successfully pupate (Figure 8).

The increase in the expression of genes for glutharedoxin, thioredoxin and HSP70 proteins, especially in the latter part of diapause, sheds light on the effects of accumulated stress at the molecular level. However, the expression of some genes, such as for metallothionein, ferritin, HSP20.1, HSP20.4 and partially for HSP90 and HSC70 proteins, was unaffected, proving that their transcription is dominantly under control of the diapause program. Interestingly, the expression of the constitutive HSP gene *hsc70* was reduced in the last two months of diapause under heat stress, which reflects impaired protein synthesis due to the lack of energy. This explains the consequent premature mortality of larvae and their inability to pupate in this period (Figure 8).

The findings presented in this study evidence that warmer winters, as a part of the looming climate changes and global warming, will affect the fragile ecophysiology of cold-adapted species in temperate and subpolar regions and will therefore also potentially reduce their numbers in natural populations and threaten biodiversity. However, the authors are aware of the limitations of the presented study which has not examined the effects of mild winter temperatures on the proteome and metabolome of diapausing larvae of the European corn borer, a perspective that should be the focus of future studies.

## Figures and Tables

**Figure 1 ijms-25-03100-f001:**
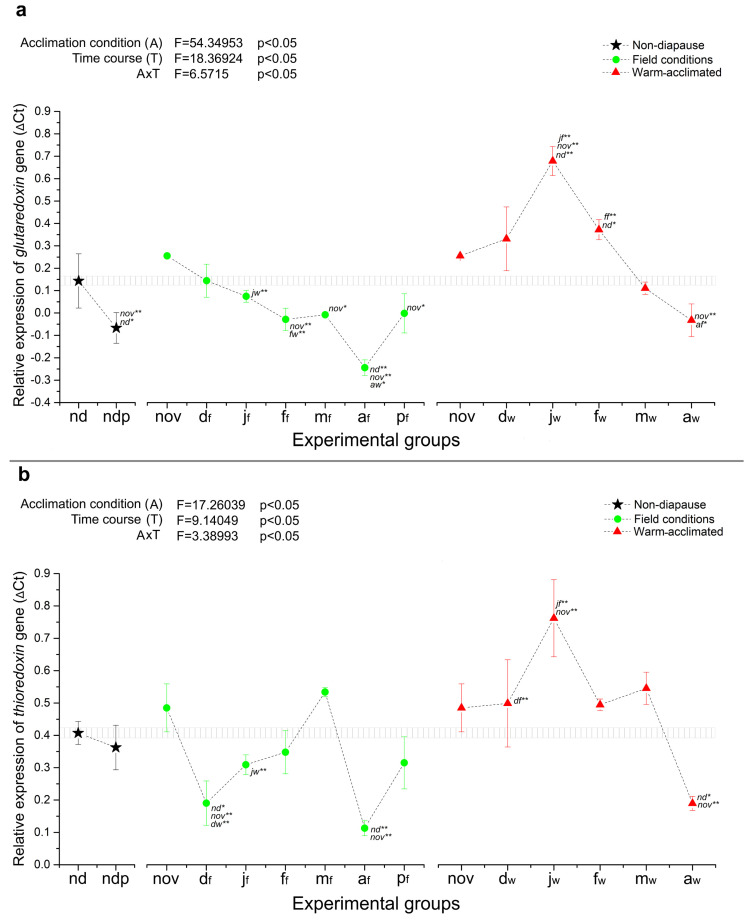
The relative expression (shown as ΔCt) of *grx* (**a**) and *trx* (**b**) in whole-body homogenates of non-diapausing larvae (nd) and pupae (ndp), as well as diapausing larvae of *O. nubilalis* held in either field (f) or warm acclimation (w) conditions from November to April (nov, d_f_, d_w_, j_f_, j_w_, f_f_, f_w_, m_f_, m_w_, a_f_, a_w_; first letter denotes month, second letter in subscript denotes acclimation condition), and pupae emerged from diapausing larvae under field conditions in April (p_f_). Ct values obtained from qRT-PCR were normalized to actin and the statistical significance of determined values was tested using one-way ANOVA followed by *post hoc* Fisher’s test, with levels of significance of *p* < 0.05 (*) and *p* < 0.01 (**). Error bars represent the standard error of the mean of three biological replicates for qRT-PCR results. Significant differences in expression levels compared to nd (non-diapause) and nov (November) are marked as nd and nov, respectively. Significant differences between the same months under different experimental conditions are also marked. Interaction between acclimation conditions (A) and time course (T) was analyzed in groups of diapausing larvae using two-way ANOVA followed by *post hoc* Tukey’s test, with a level of significance of *p* < 0.05 (upper left). Here, if the *p*-value (probability) is greater than 0.05, there is no statistical significance.

**Figure 2 ijms-25-03100-f002:**
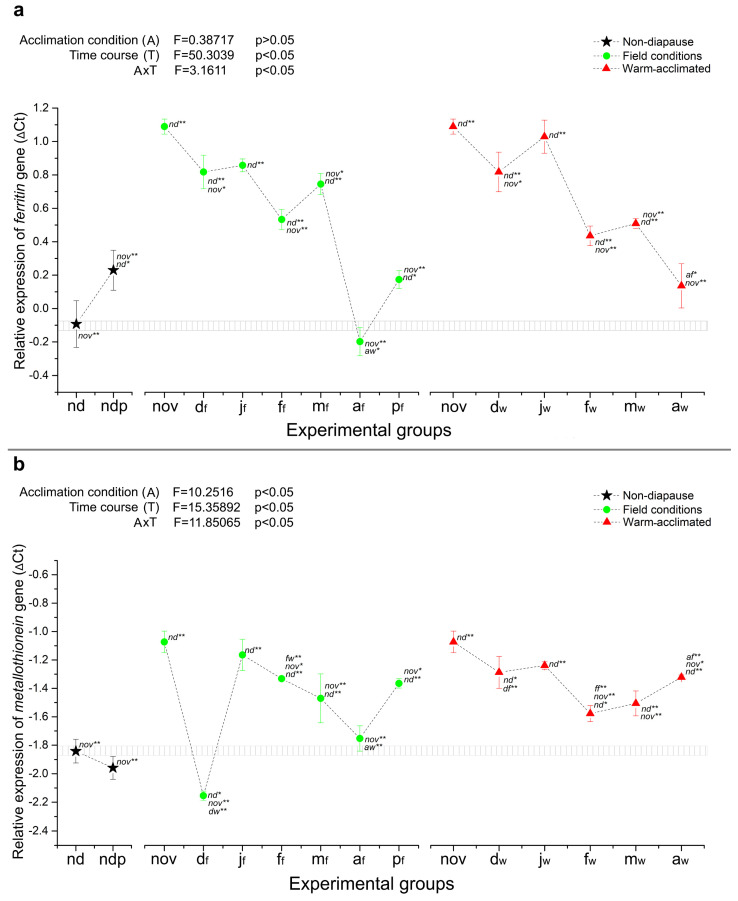
The relative expression (shown as ΔCt) of *fer* (**a**) and *mtn* (**b**) in whole-body homogenates of non-diapausing larvae (nd) and pupae (ndp), as well as diapausing larvae of *O. nubilalis* held in either field (f) or warm acclimation (w) conditions from November to April (nov, d_f_, d_w_, j_f_, j_w_, f_f_, f_w_, m_f_, m_w_, a_f_, a_w_; first letter denotes month, second letter in subscript denotes acclimation condition), and pupae emerged from diapausing larvae under field conditions in April (p_f_). Statistical significance of the results was tested using one-way ANOVA followed by *post hoc* Fisher’s test, with levels of significance of *p* < 0.05 (*) and *p* < 0.01 (**). All other statistical analysis was performed as described for Figure 1.

**Figure 3 ijms-25-03100-f003:**
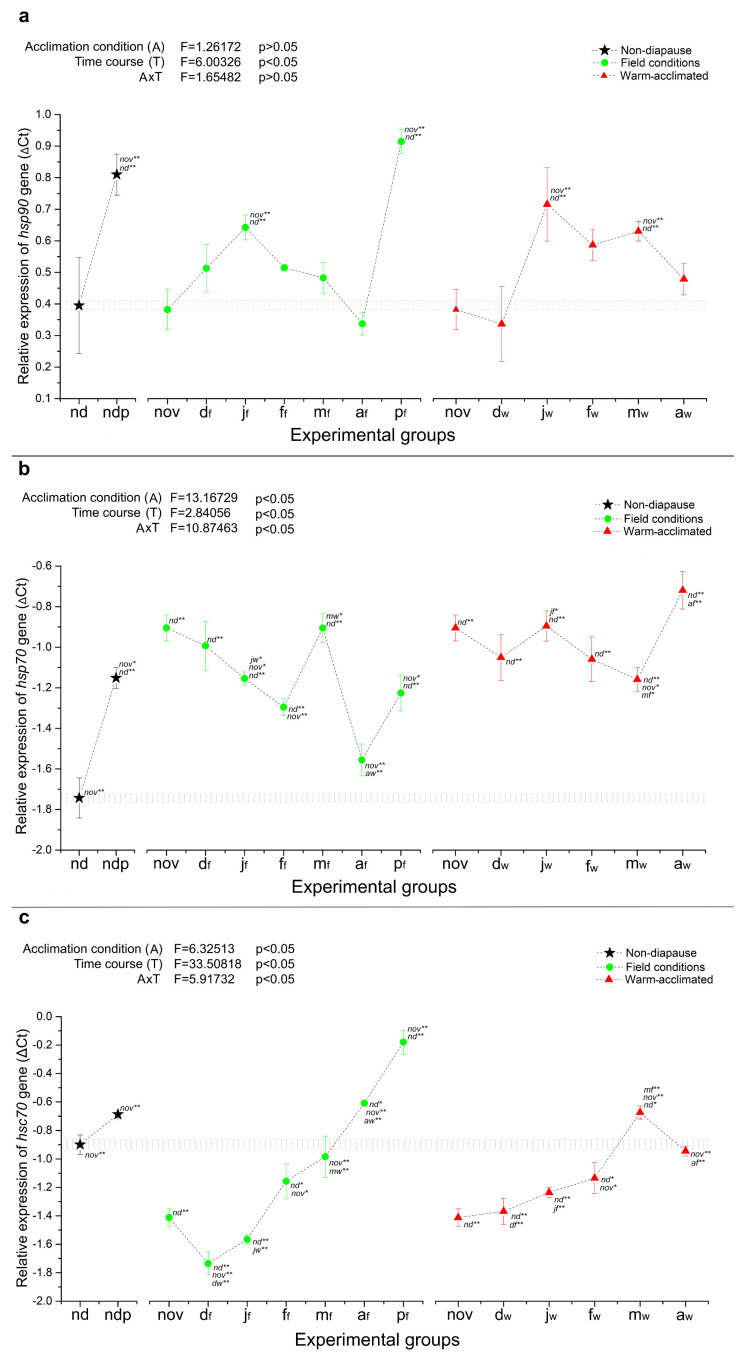
The relative expression (shown as ΔCt) of *hsp90* (**a**), *hsp70* (**b**) and *hsc70* (**c**) in whole-body homogenates of non-diapausing larvae (nd) and pupae (ndp), as well as diapausing larvae of *O. nubilalis* held in either field (f) or warm acclimation (w) conditions from November to April (nov, d_f_, d_w_, j_f_, j_w_, f_f_, f_w_, m_f_, m_w_, a_f_, a_w_; first letter denotes month, second letter in subscript denotes acclimation condition), and pupae emerged from diapausing larvae under field conditions in April (p_f_). Statistical significance of the results was tested using one-way ANOVA followed by *post hoc* Fisher’s test, with levels of significance of *p* < 0.05 (*) and *p* < 0.01 (**). All other statistical analysis was performed as described for Figure 1.

**Figure 4 ijms-25-03100-f004:**
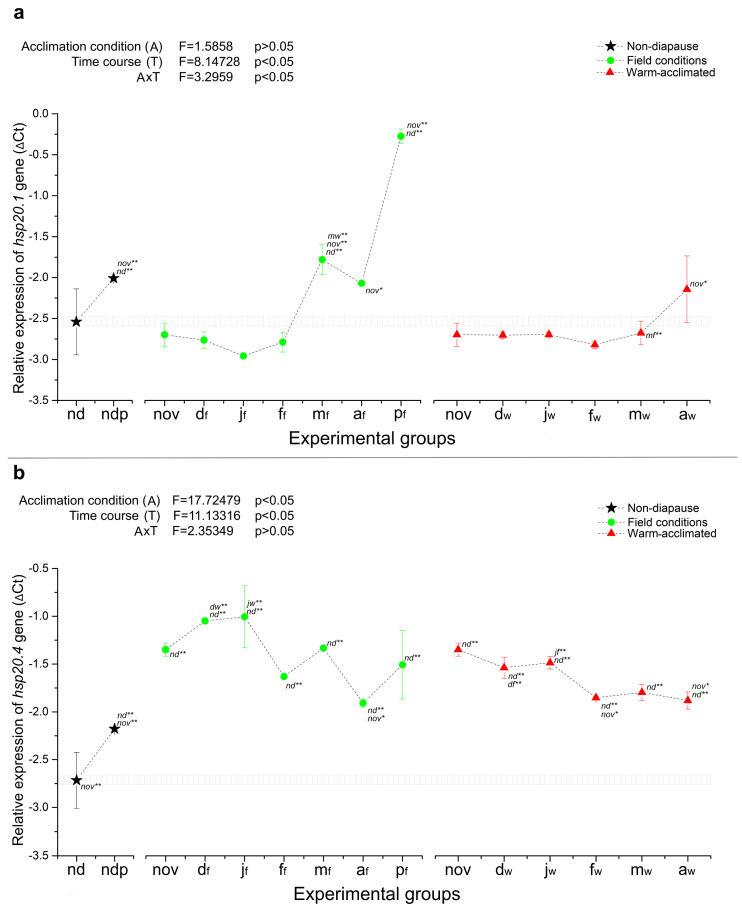
The relative expression (shown as ΔCt) of *hsp20.1* (**a**) and *hsp20.4* (**b**) in whole-body homogenates of non-diapausing larvae (nd) and pupae (ndp), as well as diapausing larvae of *O. nubilalis* held in either field (f) or warm acclimation (w) conditions from November to April (nov, d_f_, d_w_, j_f_, j_w_, f_f_, f_w_, m_f_, m_w_, a_f_, a_w_; first letter denotes month, second letter in subscript denotes acclimation condition), and pupae emerged from diapausing larvae under field conditions in April (p_f_). Statistical significance of the results was tested using one-way ANOVA followed by *post hoc* Fisher’s test, with levels of significance of *p* < 0.05 (*) and *p* < 0.01 (**). All other statistical analysis was performed as described for Figure 1.

**Figure 5 ijms-25-03100-f005:**
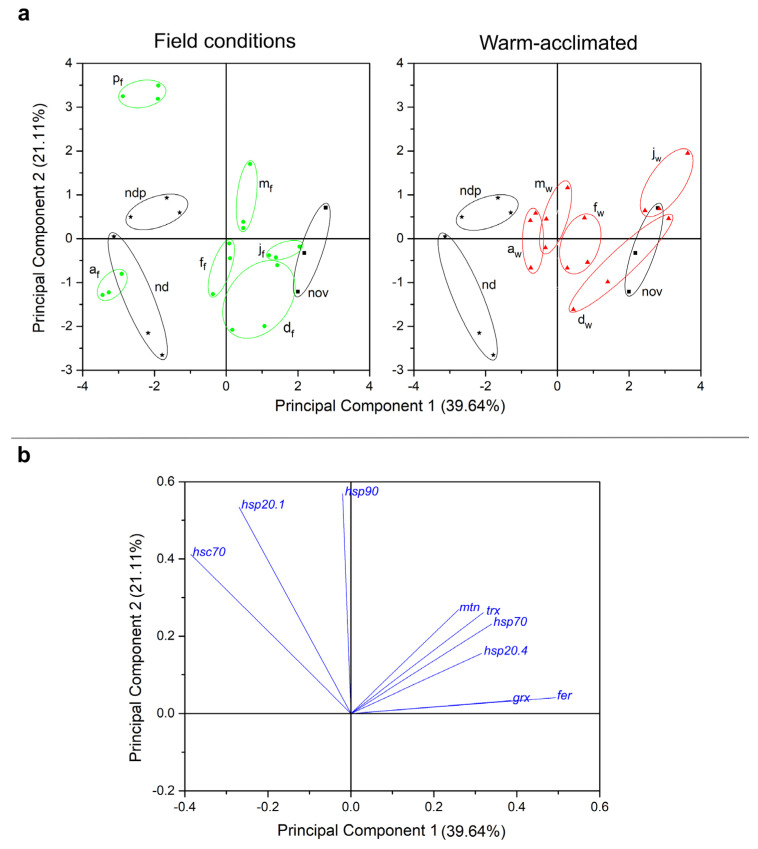
Plots of component scores (symbols; (**a**)) and of the variable loadings (vectors; (**b**)) for the two principal components from PCA performed on the relative expression of analyzed genes in whole-body homogenates of non-diapausing larvae (nd) and pupae (ndp), as well as diapausing larvae of *O. nubilalis* held in either field (f) or warm acclimation (w) conditions from November to April (nov, d_f_, d_w_, j_f_, j_w_, f_f_, f_w_, m_f_, m_w_, a_f_, a_w_; first letter denotes month, second letter in subscript denotes acclimation condition), and pupae emerged from diapausing larvae under field conditions in April (p_f_). Black stars denote nd and ndp groups, black squares denote the nov group, green circles denote field-acclimated groups and red triangles denote warm-acclimated groups. Scores are scaled by the square root of the eigenvalues (i.e., scaling = 2), and all samples consisted of three independent groups (three larvae/pupae per group).

**Figure 6 ijms-25-03100-f006:**
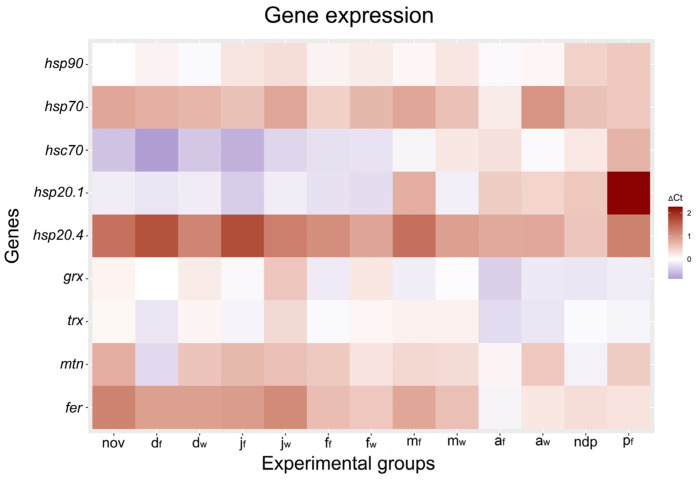
Relative expression heat map for all analyzed genes, normalized to *actin*. Experimental groups: diapausing larvae of *O. nubilalis* held in either field (f) or warm acclimation (w) conditions from November to April (nov, d_f_, d_w_, j_f_, j_w_, f_f_, f_w_, m_f_, m_w_, a_f_, a_w_; first letter denotes month, second letter in subscript denotes acclimation condition); non-diapausing pupae (ndp) and pupae emerged from diapausing larvae under field conditions in April (p_f_). Different colors on the righthand scale indicate increased (positive values) or decreased (negative values) relative expression when compared with non-diapausing larvae (nd, control, value 0). The heat map was created in R Statistical Software version 4.3.2 (https://www.r-project.org/ accessed on 20 December 2023).

**Figure 7 ijms-25-03100-f007:**
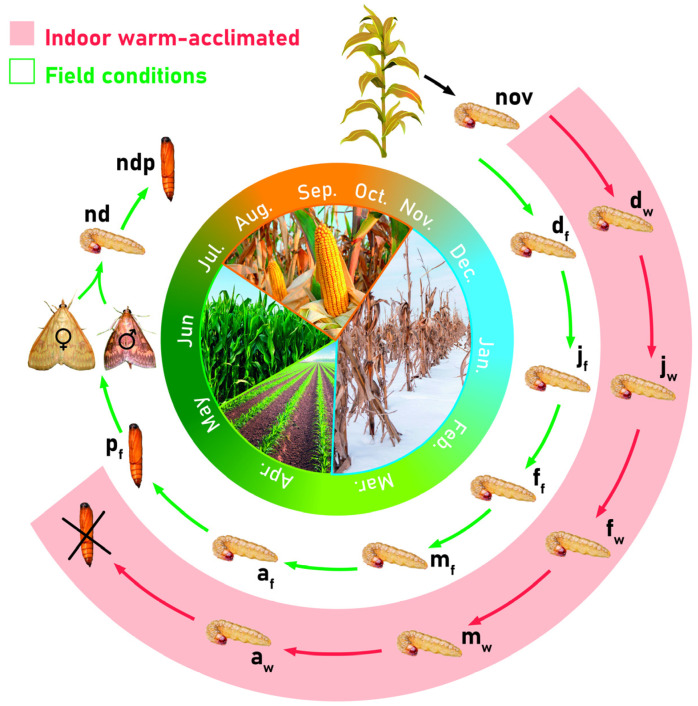
Circular illustration of the experiment. Explanations given in the text.

**Figure 8 ijms-25-03100-f008:**
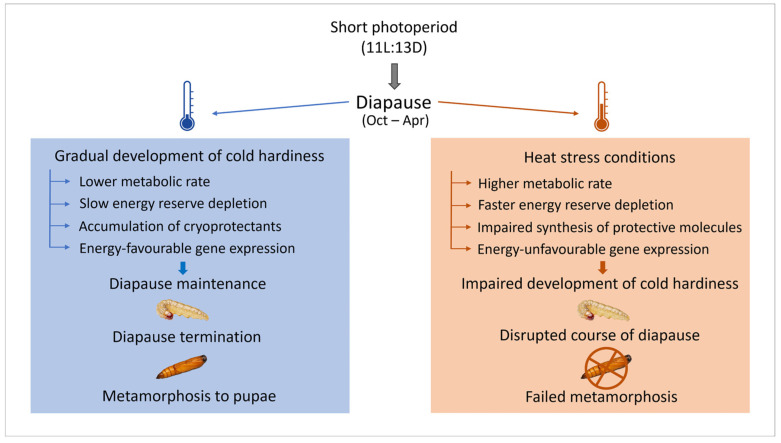
The influence of low and high winter temperatures on the metabolism, gene expression and survival of diapausing larvae of *Ostrinia nubilalis*.

**Table 1 ijms-25-03100-t001:** Composition of artificial diet used for maintaining non-diapausing larvae.

Component	Amount (g)	Component	Amount (g)
Wheat kernels	160.0	Citric acid	5.0
Barley kernels	25.0	Vitamin C	5.0
Brewer’s yeast	25.0	Methylparaben	1.7
Agar	20.0	Acetic acid	1.4
Formaldehyde 37%	6.0	Vitamin B complex	0.07

**Table 2 ijms-25-03100-t002:** Composition of individual reaction mixtures for qRT-PCR analysis.

Component	Volume (μL)
2X Power SYBR™ Green PCR Master Mix	7.0
F primer (10 μM)	0.7
R primer (10 μM)	0.7
DEPC-treated H2O	0.6
cDNA mixture (~12.5 ng/μL)	5.0
Total per reaction	14.0

**Table 3 ijms-25-03100-t003:** Oligonucleotides used in the qRT-PCR (F, forward; R, reverse; Tm, temperature of melting).

Gene	Primer Sequence (5′-3′)	Tm (°C)	Accession No.
*Actin* (reference)	F: CAGAAGGAAATCACAGCTCTAGCC R: ATCGTACTCCTGTTTCGAGATCCA	63.33 62.95	EL928709.1
*Glutaredoxin* (*grx*)	F: TCGGCAAGGTCAAACAACCA R: GCCACCTCCTACACAGTTCC	57.50 57.60	EL930102.1
*Thioredoxin* (*trx*)	F: GGGTTTTGATGACACTGACGC R: ACGCTTCTACGGTGACAACA	56.70 56.50	EL929289.1
*Ferritin* (*fer*)	F: GGCGCTCACTTCTCTAAGGATACTG R: GATTAGTGAGGTGACGTCAGAGGTG	63.82 63.46	EL929400.1
*Metallothionein* (*mtn*)	F: AAAGAGACACAGCTCCTCCAAATTC R: ATTGAGACACAGCCACTTCATCTTC	63.31 63.04	EL929052.1
*Heat shock protein 90* (*hsp90*)	F: CAAGATCGTTCTTCACATCAAGGAG R: CGTCCTCTTTCTTCTCTTCTTCAGC	63.43 63.31	EL929806.1
*Heat shock protein 70* (*hsp70*)	F: GCACAGGCCGCAGCAAGAAC R: AGGGCTTGTCGCACGCTGAA	65.14 65.12	XM_028309302
*Heat shock protein cognate 70* (*hsc70*)	F: GAGGCGGAAGATTACAAGAAACAAA R: GAGATCACATTTTGCTTCAATCACG	63.78 64.08	EL928755.1
*Heat shock protein 20.1* (*hsp20.1*)	F: CAGCGCTAAAGAATGAAAGGTCTGT R: TAGGTATCTCTCATTTCGCCTGTCC	64.11 63.90	AB568468.1
*Heat shock protein 20.4* (*hsp20.4*)	F: CGAAGAAAGTATCAGACGTGTCCAA R: TAAATGCAACGCATCACGAGATTAC	63.76 64.27	AB568467.1

## Data Availability

The data presented in this study are available from the corresponding author upon reasonable request. The data are currently not publicly available due to the policy of the University of Novi Sad where PhD candidates are obliged to upload their data to the publicly available platforms of the University of Novi Sad upon completion of their PhD thesis: (https://open.uns.ac.rs/ and https://cris.uns.ac.rs/searchDissertations.jsf).

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
