# Peer review of "Prolonged Heat Stress during Winter Diapause Alters the Expression of Stress-Response Genes in Ostrinia nubilalis (Hbn.)"

_ijms, 2024, doi:10.3390/ijms25063100_

Round 1

Reviewer 1 Report

Comments and Suggestions for Authors

I read the manuscript (ijms-2880121) entitled "Prolonged heat stress during winter diapause alters the expression of stress-response genes in Ostrinia nubilalis (Hbn.)," written by Uzelac et al. for publication in the IJMS. They investigated the combined effects of thermal stress and diapause program on the expression of selected genes involved in antioxidant defence and heat shock response in the European corn borer. This is a well-written manuscript that I really enjoyed reading it. Please see my suggestions below.    

Introduction; Lines 62-63: Please provide the specific example or area.

Results; Sections 2.3 and 2.4: ANOVA and PCA analysis; please provide the details of how you performed these tests in the Statistical Analysis section in Materials Methods and discuss these results under the different subheadings rather than Two-way ANOVA and Principal Component Analysis (PCA).

Table 3: Please provide the references for each primer if they are not from this work.  

Reviewer 2 Report

Comments and Suggestions for Authors

This study stives to shed more light on the diapause of the European corn borer by taking advantage of transcriptomics, more specifically, the expression patterns of selected genes involved in homeostasis and defense mechanisms. The rationale is well-defined and the experimental takes advantage of appropriate molecular techniques.  The data are novel and adequately discussed. 

I have only a number of minor comments:

- The genetic approach provides interesting, yet only partial data to the whole picture. The interpretation of the results could be supported by protein validation using Western blots for example.

- Although interesting, the Discussion is very long. I would recommend reducing it and stick to a shorter and more compact outline.

- The discussion could also benefit from a scheme that would depict the involvement of the studied genes and their observed dynamics on the physiological processes occurring in the larvae.

- Finally, the authors should briefly discuss any limitations their study may have been impacted by and their implications on future research.

Reviewer 3 Report

Comments and Suggestions for Authors

Article by authors

Iva Uzelac , Miloš Avramov , Teodora Knežić , Vanja Tatić , Snežana Gošić-Dondo and Željko D. Popović "Prolonged heat stress during winter diapause alters the expression of stress-response genes in Ostrinia nubilalis (Hbn.)"

is well relevant, well structured, the results are clearly presented.

 There are a number of small comments:

-It seems to me that it is necessary to emphasize in the introduction the ecological component of the influence of an increase in temperature on the development of such an important component of the environment as insects, using the example of your research object.

-L 115 check the repeated terms “...hsc70, hsc70....”

-L 128 decode the abbreviation “nd” in the test in the captions to Figures 1-4, it is necessary to specify (decipher) all abbreviations on the abscissa axis, the same abbreviations must be deciphered in the text at the first mention

-L 205 is the word parameters missing? After the phrase “between two"?

-L 262 “The presented here is the first of its kind to be 262 conducted on this important pest of corn.” An unsuccessful sentence, please rephrase it.
